# Enhanced Tribological and Mechanical Properties of Copper-Modified Basalt-Reinforced Epoxy Composites

**DOI:** 10.3390/polym17010091

**Published:** 2025-01-01

**Authors:** Corina Birleanu, Mircea Cioaza, Razvan Udroiu, Marius Pustan, Paul Bere, Lucian Lazarescu

**Affiliations:** 1MicroNano Systems Laboratory, Mechanical Systems Engineering Department, Technical University from Cluj-Napoca, Blv. Muncii nr. 103-105, 400641 Cluj-Napoca, Romania; corina.barleanu@omt.utcluj.ro (C.B.); marius.pustan@omt.utcluj.ro (M.P.); 2Manufacturing Engineering Department, Transilvania University of Brasov, Blv. Eroilor nr. 29, 500036 Brașov, Romania; 3Department of Manufacturing Engineering, Technical University from Cluj-Napoca, 400001 Cluj-Napoca, Romania; paul.bere@tcm.utcluj.ro (P.B.); lucian.lazarescu@tcm.utcluj.ro (L.L.)

**Keywords:** basalt fiber/epoxy composite material, tensile strength, friction coefficient COF, dry abrasion wear, wear rate, cuprous powder

## Abstract

The increasing demand for high-performance materials in industrial applications highlights the need for composites with enhanced mechanical and tribological properties. Basalt fiber-reinforced polymers (BFRP) are promising materials due to their superior strength-to-weight ratio and environmental benefits, yet their wear resistance and tensile performance often require further optimization. This study examines how adding copper (Cu) powder to epoxy resin influences the mechanical and tribological properties of BFRP composites. Epoxy matrices, modified with 5%, 10%, and 15% weight fractions (wf.%) of copper powder, were reinforced with BFRP-type fabric, using a vacuum bag manufacturing method. Mechanical tests, including bending and tensile tests, showed notable improvements in tensile strength and flexural modulus due to copper addition, with higher copper (Cu) content enhancing ductility. Tribological tests using a pin-on-disk tribometer revealed reduced wear rates and an optimized coefficient of friction. Statistical analysis and 3D microscopy identified wear mechanisms such as delamination and protective copper film formation. The results highlight the significant potential of copper-modified BFRP composites for applications demanding superior mechanical and tribological performance.

## 1. Introduction

Technological advancements have fueled the demand for high-performance materials tailored to specific industrial applications, driving rapid growth in the development of composite materials. Among these, polymer matrix composites (PMCs) stand out due to their superior mechanical strength, chemical resistance, and versatility. Basalt fiber-reinforced polymers (BFRPs) have garnered significant attention for their high strength-to-weight ratio, thermal stability, and environmental benefits, making them a cost-effective alternative to traditional carbon or glass fiber composites [1,2,3].

The mechanical and tribological performance of BFRPs depend significantly on matrix modifications and reinforcement techniques. For instance, basalt fibers have demonstrated enhanced properties when treated with silane-based adhesion promoters or silica nanoparticles, improving tensile and interlaminar shear strength (ILSS) [4,5,6]. Similarly, surface treatments, such as plasma or acid etching, have been shown to improve fiber-matrix adhesion and fracture toughness [7,8]. Despite these advancements, BFRPs often require further optimization for applications involving high wear and friction [9,10,11].

Recent studies have explored the addition of metallic fillers to polymer matrices to enhance wear resistance and thermal conductivity. Copper powder, in particular, has been identified as a promising additive due to its tribological and mechanical properties. Research indicates that incorporating metallic particles can significantly reduce wear rates and improve friction coefficients by forming protective layers on contact surfaces [12,13,14].

The flexural energy values of BFRP have been reported to vary significantly in previous studies. Subagia et al. [10] documented values exceeding 220 MPa, while Lopresto et al. [11] observed values around 500 MPa. Similarly, flexural modulus values range widely, from over 4 GPa, as noted by Bulut [12], to more than 20 GPa, as reported by Lopresto et al. [11].

Interlaminar shear strength (ILSS) has also been extensively studied, revealing dependence on the manufacturing method. For example, Lopresto et al. [11] found ILSS values of approximately 18 MPa for basalt FRC produced using vacuum-assisted resin infusion (VARI) with a hand roller method. The ILSS increased to around 40 MPa for composites made with vacuum-assisted resin transfer molding (VARTM). Bi-directional woven basalt/epoxy composites were reported to achieve ILSS values exceeding 55 MPa [11,13,14,15]. Additionally, Rybin et al. [16] demonstrated that applying a dense zirconia coating on the fiber surface significantly slowed the corrosion of basalt fibers in alkaline solutions.

Basalt fiber composites have also been evaluated for their tribological properties. For instance, low-density polyethylene-based composites reinforced with basalt fillers exhibited wear rates that increased with higher loads and sliding speeds, with wear being significantly influenced by basalt content [17,18,19]. Similarly, Vannan et al. [20,21] highlighted that short basalt fibers in aluminum–metal matrix composites effectively reduced the friction coefficient. These findings underscore the potential of basalt fiber composites in applications requiring superior mechanical and tribological performance.

Wang et al. [21,22] further explored basalt fiber reinforcement in polyether–ether–ketone (PEEK) composites. They reported significant improvements in tensile strength and reductions in the specific wear rate with increased basalt fiber content, demonstrating the suitability of these materials for high-performance engineering applications.

This study focuses on the impact of copper powder modification on the mechanical and tribological properties of BFRP composites. Using vacuum bag technology, epoxy matrices were modified with 5%, 10%, and 15% copper content (by weight) and reinforced with basalt fiber fabric to produce composite laminates. Tensile and bending tests, as well as dry sliding wear tests conducted according to ASTM standards, were analyzed. Statistical evaluations, including ANOVA, assessed the effects of copper content, applied load, and sliding speed on tribological performance. Wear mechanisms were examined through 3D microscopy, revealing delamination and the formation of protective copper films.

The findings provide insights into how copper powder modifications enhance the strength, flexibility, and wear resistance of basalt fiber composites, contributing valuable data for applications in high-wear environments. These copper-modified BFRP composites show promising potential for various industrial applications where both mechanical strength and wear resistance are crucial. Specifically, they could be valuable in automotive components such as brake pads, clutch plates, and bearing materials where tribological properties are paramount. In aerospace applications, these materials could be utilized in landing gear components and structural elements where high strength-to-weight ratios combined with wear resistance are essential. Moreover, in marine engineering, these composites could be particularly suitable for propeller shafts, bearings, and other components exposed to harsh environmental conditions, as they combine the corrosion resistance of basalt fibers with the enhanced wear resistance provided by copper modification. The mining and material handling industries could also benefit from these materials in conveyor systems, chutes, and wear plates where both impact resistance and sliding wear resistance are required.

While previous studies have explored various modifications of BFRP composites, the systematic investigation of copper powder’s influence on both mechanical and tribological properties represents a novel approach in this field. This research uniquely combines vacuum bag manufacturing with precise copper powder incorporation at different weight fractions, addressing a significant gap in the literature. Furthermore, the comprehensive analysis of wear mechanisms through statistical methods and 3D microscopy offers new insights into the formation of protective copper films during wear processes, contributing to the fundamental understanding of these hybrid composites.

## 2. Materials, Samples Manufacturing and Properties

### 2.1. Basalt Fiber Reinforced Polymer Composites with Cooper Powder (BFRP+Cu)

The manufacturing of the 50 wf.% basalt fiber-reinforced composite material involved a meticulously controlled process to produce high-quality, standardized test specimens. HEXION GmbH (Duisburg, Germany) provided the EPIKOTETM MGS LR 135 epoxy resin system, which served as the basis for the polymer matrix. It was mixed with the curing agent EPIKURETM MGS LH136 at a precise weight ratio of 100:35. To investigate the effects of metal reinforcement, the epoxy matrix was further modified with copper (Cu) powder, added in varying concentrations of 5%, 10%, and 15% by weight, aiming to enhance the composite’s mechanical and tribological properties.

A basalt fiber fabric (220 g/m^2^, twill fabric) was selected as the reinforcing material for its superior strength-to-weight ratio, thermal stability, and excellent polymer compatibility, making it ideal for demanding applications. Composite fabrication involved five layers of basalt fiber fabric (500 × 300 mm) stacking sequence applied in a metallic plate mold. Hand lay-up impregnation using the wet lay-up technique ensured uniform distribution and thorough fiber saturation with the epoxy mixture.

Following impregnation, the layered material was enclosed in a vacuum bag. This vacuum-assisted process was critical for removing air bubbles and excess resin, resulting in a void-free and uniform composite structure. The vacuum-sealed material underwent a controlled autoclave curing cycle, which provided the exact heat and pressure conditions necessary for the complete cross-linking of the epoxy matrix. This step significantly enhanced the composite’s mechanical strength, durability, and dimensional stability.

The autoclave cycle was 180 min at 120 °C, with a vacuum bag pressure of −0.9 bar, internal pressure of 4 bar in the autoclave, and cooling for 60 min down to 30 °C.

After the autoclave curing procedure, the process yielded three composite plates, each with dimensions of 500 × 300 × 1 mm. These plates were precisely cut using laser cutting technology to produce standardized test specimens. The following specimen types were prepared:

Tribological test disks: diameter, 50 mm; thickness, 1 mm; for analyzing wear resistance.

Tensile test specimens: dimensions of 250 × 25 × 1 mm, prepared according to ASTM D3039-17.

Bending test specimens: dimensions of 80 × 13 × 1 mm, prepared in compliance with ASTM D7264D.

Adhering to these standards ensured the specimens were of consistent quality, enabling accurate and reproducible mechanical testing.

In conclusion, this carefully designed manufacturing process—encompassing controlled reinforcement, copper powder modification, and standardized specimen preparation—was aimed at producing high-quality composite materials. This approach facilitates a comprehensive evaluation of the influence of basalt fibers and copper on the composite’s performance characteristics, paving the way for advancements in high-performance engineering applications such as for automotive, aerospace, and structural components.

Previous research has shown that ILSS values for basalt fiber composites can vary significantly depending on the manufacturing technique. For instance, composites manufactured using vacuum-assisted resin infusion (VARI) demonstrate ILSS values of approximately 18 MPa, while those made with vacuum-assisted resin transfer molding (VARTM) can reach up to 40 MPa, as reported by Lopresto et al. [11]. This highlights the importance of selecting an appropriate fabrication method to achieve desired mechanical properties. These variations underscore the need to align manufacturing techniques with application requirements. By combining optimized manufacturing methods with material modifications, such as copper addition, further improvements in performance could be achieved.

Table 1 shows the composition of the prepared laminates, detailing the weight percentages (wf.%) of the matrix and reinforcements, as well as the designation of each laminate.

### 2.2. The Ball Sample

For the tribological tests, a bearing ball made of 52100-quality carbon steel alloyed with chromium was used as a counter sample. The standard balls utilized in the test possess a diameter of 12.7 mm. These balls, widely employed in industrial applications as ball or roller bearings, are recognized for their exceptional surface finish, outstanding wear resistance, high hardness, and superior load-bearing capacity. Table 2 details the chemical composition and mechanical properties of the balls used in this study, with the data provided by the supplier. RKB Bearing Industries Group, Switzerland, a key player in the bearing industry since 1936, supplied the balls. Drawing on decades of experience, RKB has the specialized knowledge and skills required to design and manufacture industrial bearings from 1 mm inside diameter to over 2000 mm outside diameter.

## 3. Experimental Method and Device

### 3.1. Evaluation of Mechanical Properties of BFRP Samples

The performance of the laminates under external loading was assessed through a series of mechanical property tests. All tests were conducted at ambient pressure and temperature in a controlled laboratory environment, adhering to ASTM and/or ISO standards. Specimens, cut to the dimensions specified by these standards, were prepared using a waterjet cutting machine. This method was chosen to prevent the thermal stress accumulation that typically results from heat generated by conventional band saw or diamond-cutting units.

Five specimens of the BFRP materials were tested for their tensile and flexural properties following standard protocols ASTM D638-14 (Standard Test Method for Tensile Properties of Plastics) and ASTM D7264D (Standard Test Methods for Flexural Properties of Unreinforced and Reinforced Plastics). Flexural testing was performed using a three-point bending setup on a Zwick Roell Z150 universal testing machine—The ZwickRoell Group from Ulm, Germany (Figure 1). The mechanical tests, including tensile and bending tests, were conducted using the Zwick Roell Z150 universal testing machine. This high-precision equipment, with a maximum load capacity of 150 kN, is specifically designed for a wide range of mechanical testing applications. The Z150 is equipped with state-of-the-art load cells and displacement sensors, ensuring high accuracy and repeatability in the measurement of forces and deformations. The machine’s rigid frame construction minimizes system compliance, making it particularly suitable for materials requiring stringent testing conditions.

The Zwick Roell Z150 operates with integrated software, enabling precise control over test parameters, real-time data acquisition, and advanced post-test analysis. For tensile tests, the specimens were clamped using specialized grips designed to prevent slippage or stress concentration effects. In bending tests, a three-point bending fixture was employed, ensuring consistent loading and accurate characterization of material flexural properties.

The tests adhered to international standards, validating the Z150 as a reliable and versatile platform for generating scientifically robust data, which was subsequently utilized for this study’s analysis and conclusions.

Each specimen was loaded at a test speed of 2 mm/min until failure, and the average values of the flexural mechanical properties, along with the standard deviations across all specimens, were collected.

Tensile specimens were tested on the same machine, with strain data collected via transverse displacement (Figure 1b). The stress–strain curves for flexure and tensile tests are displayed, and the average tensile mechanical properties, along with standard deviations for the specimens tested, were collected.

### 3.2. Pin-on-Disk Test

The characterization of friction and wear properties, including wear rates and wear resistance, is commonly conducted using various types of tribometers, with the pin-on-disk test being among the most widely employed methods. The popularity of this technique stems from its relative simplicity and its applicability to a broad range of tribological contacts, which can be effectively modeled by the motion of a pin on a disk. Experimental procedures for such tests are standardized and adhere to recognized protocols, including ASTM G99 (Standard Test Method for Wear Testing with a Pin-on-Disk Apparatus) and ASTM F732 (Standard Test Method for Wear Testing of Polymeric Materials).

Schematically, the pin-on-disk test is described in Figure 2.

A fixed rod is pressed against a rotating disk under a predetermined load. While the pin can adopt various shapes, spherical (ball or lens) and cylindrical configurations are the most commonly used due to their ease of alignment. In contrast, flat pins often present challenges such as misalignment, leading to uneven loading and complications in theoretical analysis. Throughout the test, key parameters such as frictional force, wear, and temperature are continuously monitored to ensure accurate and comprehensive data acquisition.

The tribometer is calibrated periodically to ensure that the measured data are accurate and precise.

The software also includes by default the “Modelization” module that allows the simulation of contact stress and strain distribution. TRB^3^ disk pin tribometer is provided by Anton Paar GmbH, Graz, Austria (Figure 3) is compliant with ASTM G99, ASTM G133, and DIN 50324 standards, so we are always confident that we have performed tribological testing in accordance with industry best practices.

Parameters tested, as listed in Table 3, included applied force, sliding velocity, speed, humidity, temperature, contact duration, and contact type. Each of these parameters influences the tribological performance of the material and can change the friction and wear mode. The applied loads ranged are between 10, 20, and 30 N and allow the behavior under various stress regimes to be evaluated. The contact force affects the pressure on the surface, and hence friction and wear. A higher force tends to increase the wear of the composite material, also generating an increase in temperature at the contact. This moderate stress regime was chosen to allow evaluation of the wear transition and is relevant for practical applications.

Sliding velocities of 0.1, 0.25, and 0.36 m/s, covering a moderate sliding regime, were used. The sliding velocity is essential to determine the friction behavior. At higher speeds, the wear rate may increase due to greater energy dissipated by friction, although some materials may exhibit more stable tribological behavior at higher speeds due to the formation of a protective layer. The chosen sliding speed range avoids excessive heating, provides a stable friction regime and a controlled wear regime.

The test duration was 120 min. A prolonged test interval allows the observation of long-term wear effects, which are relevant for assessing the durability of the coupling in practical applications.

Dry friction conditions, without lubrication, were used to analyze the performance of the composite material in the direct contact regime. Dry friction amplifies friction and wear and is an effective method for testing the natural wear resistance of the material.

During the more than 350 tests performed, great attention was paid to rigorously controlling and monitoring the operating conditions to ensure the accuracy and relevance of the data obtained and to draw meaningful and applicable conclusions from the experimental results.

An electric motor rotated a disk that was precision-machined to an outer diameter of 50 mm and a thickness of 2 mm. The pins were replaced by 12.7 mm diameter bearing balls, which fit into a clamping system. Temperature, wear, and friction forces were continuously monitored throughout the test, with an accuracy tolerance of 2–3%. New balls and disks were used for each experiment, which were cleaned with technical cleaner and wiped dry before starting the experiment. The experiment lasted for 120 min, during which the friction coefficient was continuously recorded to identify the inlet friction region and the stationary friction coefficient. Temperature changes were also monitored throughout the test. The wear loss of the balls and disks was gravimetrically quantified and then analyzed using microscopic examination and surface profile measurements.

After a set number of cycles, the test was stopped, and a surface topography analysis was conducted to assess wear and document changes in surface roughness. Surface modifications were examined using Nano Focus Optical 3D Microscopy, with advanced μsurf technology for precise three-dimensional surface characterization. This method uses continuous focus variation with fixed focal length objectives, and the magnification was set to 10x. Radial layers from the scanned surface were isolated and studied to gain deep insights into its attributes.

#### Wear Calculation in the Ball-on-Disk Configuration

The amount of wear is calculated by measuring the relevant linear dimensions of the ball and disk specimens or by weighing them both before and after the test, in accordance with the applicable ASTM standards. The accompanying Figure 3 displays the typical ball-on-disk configuration along with the parameters needed to compute wear.

The following formula provides the disk’s volume loss assuming no appreciable ball or pin wear (from ASTM G99 Standard Test Method for Wear Testing with a Pin-on-Disk Apparatus).
(1)$$V_{\mathrm{disk}\;}=2\pi R\cdot \left(\frac{r^{2}}{\sin\frac{d}{2r}}-\frac{d}{4}\cdot \sqrt{4r^{2}-d^{2}}\right)$$
where

*R* = radius of the wear track

*d* = width of the wear track

*r* = radius of the ball

*V_disk_* = volume of wear on the disk

As an alternative, a profile scan over the disk’s wear track can also be used to identify the material loss. Debris and plastic deformation can cause variations around the wear track. To obtain a representative value, it is necessary to generate a sufficient number of profiles. The majority of profilometer software can easily determine the profile’s coverage area. In this work, since the friction loss is very low, we decided to calculate the wear rate through profilometry using Nano Focus Optical 3D Microscopy provided by NanoFocus AG, from Oberhausen, Germany.

The following equation provides the volume loss of the spherical ball (or a spherically ended pin), assuming that there is no discernible disk wear [23,24].
(2)$$V_{\mathrm{ball}\;}=\frac{\pi \cdot h}{6}\cdot \left(\frac{3\cdot d^{2}}{4}+h^{2}\right)$$
where



$h=r-\sqrt{r^{2}-\frac{d^{2}}{4}}$



*d* = diameter of the wear scar

*r* = radius of the ball

*V_ball_* = wear volume of the ball

The wear rate (K, mm^3^/Nm) determined in this study was calculated using the following equation:(3)$$K_{\mathrm{disc}\;}=\frac{V_{\mathrm{disk}\;}}{L_{\mathrm{sliding}\;}\cdot F};\;K_{\mathrm{ball}\;}=\frac{V_{\mathrm{ball}\;}}{L_{\mathrm{sliding}\;}\cdot F}$$
where

*V_disk_* = wear volume of the disk (mm^3^)

*V_ball_* = wear volume of the ball (mm^3^)

*F* = normal force (N)

*L_sliding_* = total sliding distance (m)

*K_disk_* = wear factor (wear rate) of the disk (mm^3^/m⋅N)

*K_ball_* = wear factor (wear rate) of the ball (mm^3^/m⋅N).

To reduce data variability, three sliding tests were conducted for each trial in this study, and the average values of the friction coefficient (µ) and wear rate were used for statistical analysis.

A 3D optical profilometer, as shown in Figure 4, was used to characterize the wear track profiles. The wear volume of the specimens was calculated by measuring the wear width and depth with the profilometer, and these values were subsequently incorporated into empirical mathematical equations. The calculations assumed an idealized spherical geometry for modeling the wear scars.

A Flir E30 infrared camera, featuring an IR pixel resolution of 160 × 120, was utilized to monitor temperature fluctuations during testing. Operating within a temperature range of −20 °C to 250 °C and offering a thermal sensitivity of 100 mK (0.1 °C or better), the camera enabled precise detection of temperature variations throughout the experimental procedures.

## 4. Experimental Results and Discussion

Analyzing experimental data for tribological and mechanical testing of reinforced composite materials involves a number of systematic steps, from processing the raw data to interpreting and comparing the results.

In order to structure the analysis of experimental data, the following steps were taken:**Organization of the raw data:** all the values obtained from the mechanical tests (e.g., tensile and flexural strength, modulus of elasticity) and tribological tests (coefficient of friction, wear rate, temperatures in the coupling) are collected, and are organized in tables for ease of access and further analysis. Values considered outliers were identified and eliminated by comparing them to the acceptable limits for each test, such as by statistical methods (mean ± 2 standard deviations).**Statistical analysis:** The mean, standard deviation, and confidence interval were calculated for each data set to obtain an overview of the variability and consistency of the results. Since we have different data sets (e.g., different materials or test conditions), we used ANOVA statistical tests to assess the significance of the observed differences.

### 4.1. Flexural Strength and E-Modulus of BFRP Laminates

The samples made for bending before and after the bending test are shown in Figure 5.

The flexural mechanical properties, along with the standard deviations across all specimens, are presented in Table 4 and Figure 6.

The impact of resin modification on the flexural bending strength of BFRP laminates was assessed using the three-point bending test. Figure 6 presents the flexural strength and modulus values obtained for all BFRP laminates. Flexural strength analysis revealed that copper-reinforced BFRP laminates exhibited improvements of 15% for BFRP5 and 12% for BFRP15, compared to the unfilled BFRP0 laminate. This enhancement emphasizes the role of copper powder in strengthening the composite structure.

This analysis highlights that BFRP5 achieves the best balance between flexural stress and stiffness, with the highest strength and only a slight reduction in modulus. BFRP10 and BFRP15 show diminishing performance, particularly in stiffness, making them potentially less suitable for applications requiring rigidity. On the other hand, BFRP0 offers a baseline reference, showing the lowest flexural stress but the highest modulus. The trade-offs between strength and stiffness should guide the material selection process based on specific application requirements.

The observed E-modulus trend highlights a steady decline in stiffness as the material composition is modified. While BFRP0 provides the highest stiffness, BFRP5 represents a good compromise between stiffness and strength. Beyond BFRP5, the increasing flexibility (lower modulus) may limit the material’s suitability for structural applications but could enhance its use in contexts requiring greater deformation tolerance. Optimizing the interaction between the reinforcement and matrix is critical to tailoring the material for its intended application.

### 4.2. Tensile Strength of BFRP Laminates

The samples made for tensile stress before and after the tensile test are shown in Figure 7.

The average tensile mechanical properties, along with the standard deviations for the tested specimens, are shown in Table 5 and Figure 8.

The assessed mechanical properties—strength, strain, and modulus of elasticity—showed high reproducibility across tests, as indicated by the standard deviation. This consistency underscores the reliability and significance of the results obtained.

Figure 8 illustrates the variation in the tensile strength of BFRP laminates as a function of the variation in the weight percent of copper powder added to the resin. Modifying the resin by adding copper powder resulted in improvements in the tensile strength of the laminate made from resin modified with 5, 10, and 15 wf.% copper powder compared to BFRP0 laminate made from pure resin. The tensile strength increases by approximately 40%, 39%, and 46% for the BFRP5, BFRP10, and BFRP15 composite laminates, respectively, compared to the BFRP0 laminate. In a fiber-reinforced polymer composite laminate, failure is mostly caused by crack development in the matrix. Once these cracks propagate and reach a critical point, they lead to extensive delamination due to fiber cracking.

The notable increase in tensile strength observed in BFRP15 laminates can be attributed to enhanced matrix stiffness from copper powder addition. Copper particles create robust bonds with the polymer matrix, facilitating efficient load transfer from the weaker matrix to the stronger fibers.

Previous studies have shown that the addition of SiO_2_ nanoparticles can increase tensile strength by approximately 30% and ILSS by 15% [8]. In comparison, the inclusion of copper powder in our composites resulted in tensile strength improvements of 40–46%, demonstrating its effectiveness as a reinforcement strategy.

The addition of copper powder significantly improved the tensile strength of the composites. Compared to the unmodified BFRP0 material, BFRP5 showed a 40.3% increase in tensile strength, while BFRP15 exhibited a maximum improvement of 46.2%. This indicates that copper addition enhances the stress transfer between the matrix and the fibers, leading to improved mechanical performance.

However, a slight decrease in the E-Modulus was observed with increasing copper content, with reductions ranging from −11.9% for BFRP5 to −0.8% for BFRP15. This suggests a trade-off between stiffness and ductility, which could be attributed to the plasticizing effect of copper particles within the matrix.

The observed improvement in mechanical properties, such as tensile and flexural strength, can be attributed to the enhanced stress transfer between the matrix and the basalt fibers. Copper particles may act as mechanical bridges, reducing stress concentrations and improving the overall load distribution within the composite.

Additionally, the ductility and thermal conductivity of copper particles facilitate a more uniform stress distribution, which prevents premature crack propagation and enhances the material’s strength. The inclusion of copper powder appears to reduce the porosity of the epoxy matrix, creating a denser and more cohesive structure, which contributes to the observed increase in mechanical performance.

At higher copper concentrations (e.g., 10–15 wf.%), particle agglomeration is likely, which introduces stress concentration points and weakens the structural integrity of the composite. The reduced stiffness observed with higher copper content may result from the disruption of the epoxy matrix continuity, as excessive particle addition creates weaker interfaces within the composite. The copper particles may also act as a plasticizing agent in the matrix, reducing its rigidity and leading to the observed decline in the elastic modulus.

### 4.3. Design of the Experiments (DOE) and Statistical Analysis

The tribological experiment was systematically designed by selecting control factors that influence three key targets: the coefficient of friction, specific wear rate, and temperature during the tribological analysis of the composite material. A comprehensive full factorial design comprising 36 factor combinations was implemented to thoroughly examine the impact of these factors on the specified targets. Weight fraction (wf.%), applied force (F), and sliding speed (v) were chosen as control factors, and were assigned 4, 3, and 3 levels, respectively (Table 6).

Statistical analysis was conducted to evaluate and characterize the effects of the control factors and their interactions on each target variable. The contributions of these factors to the observed variations in the targets were quantified using Generalized Linear Models (GLM), an advanced statistical approach that extends traditional Analysis of Variance (ANOVA). Statistical computations and model fitting were carried out using Minitab 19 software (Coventry, UK).

Significant control factors were identified by analyzing the ANOVA table, focusing on key statistical parameters such as F-values and *p*-values. The percentage contribution ratio (PC%) for each factor and their interactions were also computed. Data interpretation was further enhanced through graphical methods, including main effects plots, interaction plots, and confidence interval plots of the target variables as functions of the control factors. Finally, the assumptions underpinning ANOVA were rigorously validated in line with established guidelines from the literature [25]. We explored the effects of variable factors (e.g., temperature, applied load, speed) on material performance to understand how these can influence the practical use of composites.

Further, the wear mechanism of the specimens were analyzed post-testing using various surface analysis techniques, primarily to ascertain the hardness of the polymers in and around the wear traces and the characteristics of the transfer films on the analyzed components, both of which significantly influence the tribological behavior of the interactions between the polymers and steel material.

After data collection, data analysis was performed using the TRB^3^ V10.0.x software, specifically the TRB^3^ Disk Stud Tribometer. This program allows visualization of force–time and wear–time graphs, filtering and processing of raw data to eliminate noise and identify significant trends, and the extraction of tribological characteristics such as the coefficient of friction and wear rate.

The experimental dry sliding wear experimental results are presented in Table 7 and are the basis of discussion regarding the factors influencing wear performance, the significance of the statistical analysis, and how the results guide optimization.

Wear measurement was based on the weight loss of the ball and disk, using an analytical microbalance with an accuracy of 0.1 mg. The weight loss of the chromium alloy balls was insensitive to these experiments, and for the disk, the measure of weight loss for all test trials was quite small, increasing from 0.001 to 0.028 g with the amount of force applied and the sliding velocity.

Consequently, for wear interpretation, we relied on the results obtained by 3D optical scanning. Table 7 shows the wear rate of the composite disk range for the three sliding velocity values and the applied normal force values based on the profilometry testing data.

In order to give a visual image of the experimental results obtained with the 3D Optical Scan, Figure 9, Figure 10 and Figure 11 show some examples of the worn surface morphologies of basalt fiber reinforced composite disks with added copper powder—BFRP0, BFRP5, BFRP10 and BFRP15—under dry friction conditions, with a test duration of 120 min in contact with a ball made of chromium 52100-alloyed carbon steel under different tribological parameters, as well as the worn surface profile curve for the same samples.

As we can see in Figure 12, it was found that on the ball in dry friction with different samples of composite material reinforced with basalt fibers and added copper powder that the ball wear is almost negligible in all test conditions. The contacting surfaces develop a mutual adaptation. The basalt particles and copper layer filled the microscopic voids, forming a protective film on the steel ball.

Composite systems with additions of metal powders (such as copper) and quality metal materials such as 52100 steel create a favorable combination for dry friction. Moderate friction and efficient heat dissipation reduce the risk of severe wear.

The negligible wear observed on the steel ball can be attributed to a synergy of the composite material’s properties (hardness, tribological influence of copper, minimal transfer of composite material to the steel ball forming a protective layer) and the inherent resistance of 52100 steel to extreme friction conditions. The formed system ensures optimal interaction between the surfaces, minimizing damage. Therefore, we will not deal with the wear of the steel ball in the data interpretation.

The results of the statistical analysis for the target variables, coefficient of friction, specific wear rate, and temperature, are presented in Table 8, Table 9 and Table 10. A significant influence of the control factors on the targets was observed when the *p*-value was lower than the significance level of 0.05.

The Cu powder weight fraction (52.43%) and sliding speed (21.21%) have shown the highest percentage contributions to the coefficient of friction. The applied load factor and the interactions between factors had lower percentage contributions, as shown in Table 8.

The graphical results of statistical analysis are explained based on the following diagrams: the main effects plot, interaction plot, and interval plots. The main effects of the coefficient of friction, meaning the maximum mean values, were the copper powder weight fraction (wf_Cu) at level 3 (10%), the force at level 2 (20 N), and the sliding speed at level 1 (0.1 m/s), as shown in Figure 13. It was observed that there was a small difference between the level 3 and 4 for wf%, and level 2 and 3 for F.

The interaction plot matrix for the mean coefficient of friction factors showed intersecting lines, indicating the presence of interaction effects. Among these, the interaction between the applied force (F) and the sliding speed (v) had a significant impact on the coefficient of friction, as shown in Figure 14. Figure 15 displays interval plots with standard error bars for each factor in relation to the coefficient of friction. Notably, the differences in coefficient of friction for the weight fraction (wf) factor were significant, as the interval bar at level 1 (wf_Cu = 0%) did not overlap with the interval bars at levels 2 (wf_Cu = 5%), 3 (wf_Cu = 10%), or 4 (wf_Cu = 15%).

In contrast, the observed differences in the coefficient of friction for the applied force (F) were likely not significant, given that all interval bars overlapped (Figure 15b). A similar overlap pattern was also noted for the sliding speed (v), as presented in Figure 15c.

The means of coefficient of friction were lower at wf_Cu = 0%, F = 10 N, and v = 0.36 m/s, as shown in Figure 15.

The wf.% factor had the highest contribution of 41.29% on the specific wear factor, followed by applied load factor with 27.94%, as is shown in Table 9. The sliding speed had a lower contribution of 9.94% on the specific wear factor. The interactions between factors were not significant factors, based on their *p*-values being greater than the significance level of 0.05, as is shown in Table 8.

The applied load and the sliding speed are the most significant factors for the temperature, having contributions of 42.32% and 36.73%, respectively. The wf.% factor and all the interactions between factors had lower contributions on the temperature, as is shown in Table 10.

The R-squared values of 94.6% for coefficient of friction, 93.94% for specific wear rate and 95% for temperature, respectively, indicated that the model predicts the outcome of the dependent variable.

The results of the main effects plot for the coefficient of specific wear rate show that a wf_Cu at level 4 (15%), applied force at level 3 (30 N), and sliding speed at level 3 (0.36 m/s) had the main effects plot (Figure 16). The specific wear rate rises with an increase in the load applied, the weight fraction of Cu, and the sliding speed.

The interaction of wf_Cu with v had a significant influence on the specific wear rate factor, as shown in Figure 17. F had the lowest interaction influence on the specific wear rate factor with v, based on their parallel lines. The interval plots of each factor versus specific wear rate are shown in Figure 18. Lower means were obtained for wf_Cu = 0%, F = 10 N, and v = 0.1 m/s.

The main effects of temperature were obtained for the Copper powder weight fraction (wf.%) at level 3 (10%), the force at level 3 (30 N), and the sliding speed at level 3 (0.36 m/s), as is shown in Figure 19. A decreasing of the temperature was observed at level 3 for wf_Cu (15%).

The interaction of v (level 2 and 3) with F had a significant influence on the temperature factor, as shown in Figure 20.

The interval plots of each factor versus temperature are shown in Figure 21. Lower means were obtained for wf_Cu = 0%, F = 10 N, and v = 0.1 m/s. The differences in temperature with varying wf_Cu contents were likely not statistically significant, as all interval bars overlapped (Figure 9a). Conversely, the differences in mean temperature for both the applied force (F) and sliding speed (v) were significant, indicated by the absence of overlapping interval bars.

The generalized linear models were subsequently evaluated for model adequacy [26,27], and the resulting residuals conformed to a normal distribution (Figure 22).

Friction-induced heat can create a plastic state at the contact surface, enabling wear particles from the early stages of wear to fill the pores. Under higher operating parameters, these processes may help reduce frictional forces. Additionally, increased heat generation may lead to more reactions that are chemical between the environment and the contact surface, potentially forming a hard oxide layer. This layer can raise the coefficient of friction as it acts as a barrier to further material removal. Once the oxide layer reaches a certain thickness, it detaches from the surface as speed increases, initiating the removal of material from the contact surface. Consequently, once this speed threshold is surpassed, the friction coefficient decreases. The lowest and highest observed values for the alloy steel’s coefficient of friction were 0.28 and 0.7, depending on the test conditions and time stage.

The width of the abrasive wear track for BFRP samples exhibited a progressive increase with extended sliding time. Under high abrasive conditions, debris adherence to the metal counterface decreased. The larger asperities on the metal counterface induced deformation of the polymer surface, resulting in mechanisms such as plowing, microcutting, and the formation of distinct abrasive wear tracks. For BFRP, the primary wear mechanism during sliding friction involves the plastic deformation of the polymer matrix.

The wear of basalt fibers (BFs) in the reinforced samples is mainly caused by the abrasive action of metal asperities. Additionally, basalt debris particles work synergistically to enhance this abrasive process. BFs typically experience a reduction in diameter before fracturing into shorter lengths due to alternating stress effects. The debonding of the fibers from the polymer matrix follow this.

The wear rate coefficient (K) for BFRP sliding against chrome alloy steel and stainless steel is influenced by several factors, including the sliding speed. In general, as the sliding speed increases, the wear rate between BFRP and these metallic counterfaces decreases.

This phenomenon can be explained by the effect of sliding speed on the formation of a transfer layer. At lower sliding speeds, the interaction between the pin and disk leads to debris accumulation, promoting abrasive wear and increasing the wear rate. The debris is, on the other hand, partially expelled from the contact region and partially deposited onto the surface of the opposing material as the sliding velocity increases. This process aids in forming a transfer layer, which acts as a protective barrier, thereby reducing the wear rate. The friction layer forms through the compaction of wear debris generated during contact between the composite material and the steel surface. Copper particles in the BFRP (basalt fiber-reinforced polymer) composite play a crucial role by filling microscopic voids and creating a uniform protective layer on the contact surface. This layer reduces wear by minimizing direct abrasive interactions between the contacting surfaces. Additionally, the transfer and adhesion of these particles to the steel surface can create a barrier that optimizes tribological behavior by stabilizing the coefficient of friction and preventing accelerated wear [28,29,30,31,32].

Thus, the mechanism involves both mechanical effects (particle compaction) and chemical effects, such as the possible formation of oxide–metal layers at higher temperatures.

Previous studies have shown that basalt fiber composites exhibit enhanced wear resistance and mechanical properties when combined with various modifications. For instance, Wang et al. [22] demonstrated that increasing basalt fiber content in PEEK composites reduced the specific wear rate while improving tensile strength. Similarly, Vannan et al. [20,21] reported a reduction in the coefficient of friction when basalt fibers were used in aluminum–metal matrix composites. These findings align with the results of our study, where copper powder addition further optimized the tribological performance of basalt fiber composites. These findings, combined with insights from the literature, underscore the versatility of basalt fiber composites and the potential of material modifications, such as copper addition, to address specific application requirements [28,29,30,31,32].

The enhancement in tribological performance is likely due to the formation of a protective copper layer on the contact surfaces during wear tests. This layer minimizes direct abrasive interactions and reduces the overall wear rate. Copper’s high thermal conductivity plays a critical role in dissipating the heat generated during sliding, preventing the thermal degradation of the epoxy matrix and maintaining stable tribological performance.

At higher copper contents, the surface roughness of the composite may increase due to uneven particle distribution, which could elevate the coefficient of friction under certain conditions. Loose copper particles generated during sliding could act as abrasive debris, increasing the wear rate despite the otherwise protective role of the copper.

These variations in properties underscore the importance of optimizing the copper content to balance mechanical reinforcement and tribological performance. While moderate copper additions improve both aspects, higher concentrations may introduce adverse effects due to agglomeration and matrix disruption.

## 5. Conclusions

This study investigated the mechanical and tribological behaviors of BFRP composites with and without copper powder modification. Key findings indicate that the incorporation of 5–15% copper powder significantly enhances tensile strength, flexural strength, and wear resistance, with optimal performance observed at 5% Cu content. These enhancements suggest improved interfacial bonding and stress transfer within the composite. However, higher copper content (10–15%) leads to particle agglomeration, reducing stiffness and creating stress concentration points, which may limit its applicability in rigidity-critical applications.

The addition of Cu powder notably influences tribological properties. Lower friction coefficients were observed with specific parameters (e.g., Cu = 0%, F = 10 N, v = 0.36 m/s), while the wear rate increased with higher Cu content, load, and sliding speed. Copper’s ability to form protective layers and dissipate heat contributes to improved wear resistance, highlighting its potential for high-wear environments.

These findings align with the existing literature, underscoring the importance of optimizing copper content and processing techniques. Future research should focus on combining advanced manufacturing methods with tailored material modifications to further enhance the performance of basalt fiber composites in demanding applications.

This work demonstrates the critical role of material composition in achieving a balance between mechanical and tribological performance, offering valuable insights for designing advanced composite materials for industrial use.

## Figures and Tables

**Figure 1 polymers-17-00091-f001:**
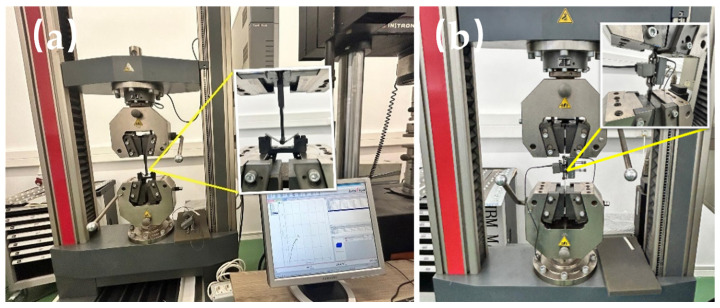
Zwick Roell Z150 universal testing machine for bending (**a**) and tensile test (**b**).

**Figure 2 polymers-17-00091-f002:**
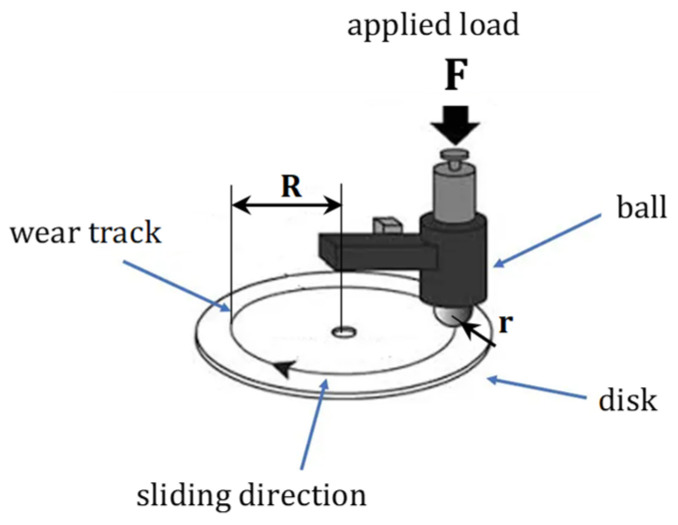
Typical ball-on-disk configuration, in which the wear track’s radius is R, the ball’s radius is r, and the normal force acting on the ball is F.

**Figure 3 polymers-17-00091-f003:**
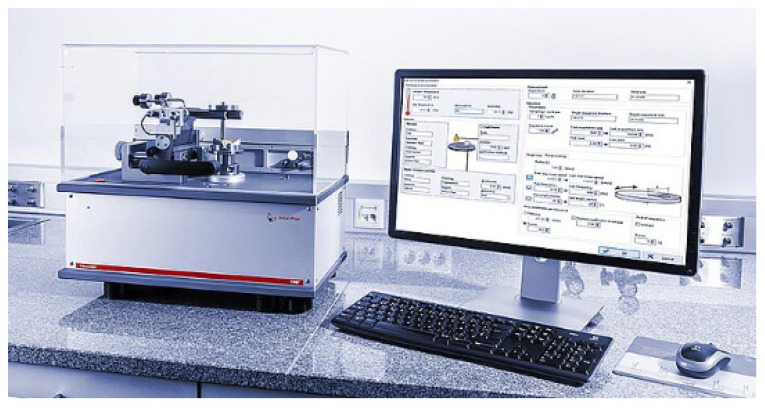
TRB^3^ disk pin tribometer.

**Figure 4 polymers-17-00091-f004:**
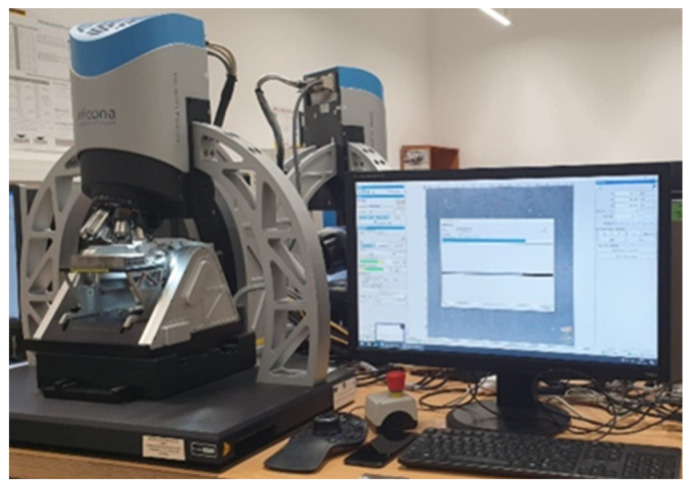
Nano Focus Optical 3D Microscope.

**Figure 5 polymers-17-00091-f005:**
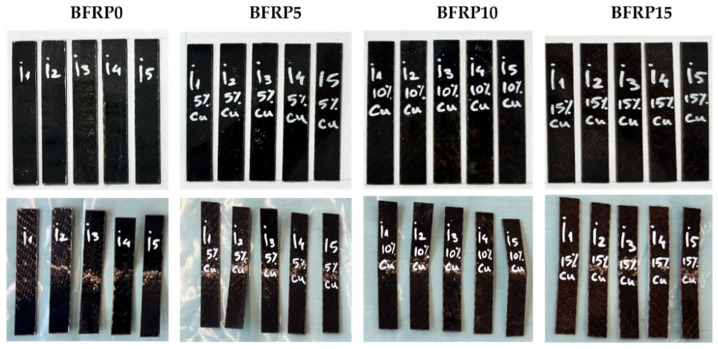
BFRP specimens before (row 1) and after bending stress (row 2).

**Figure 6 polymers-17-00091-f006:**
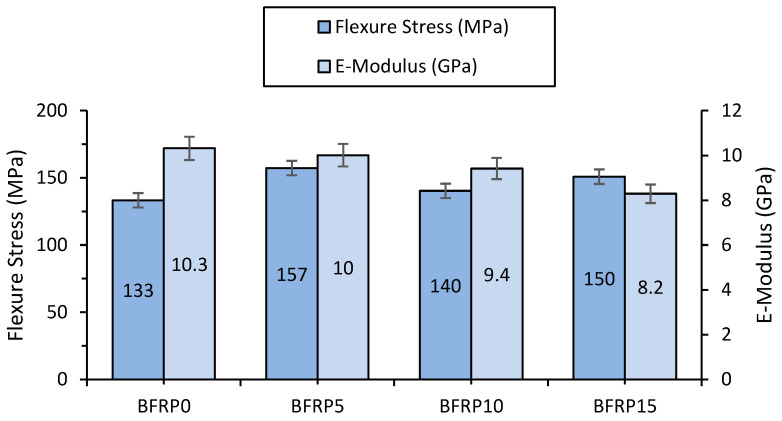
Flexural strength and modulus of BFRP laminates.

**Figure 7 polymers-17-00091-f007:**
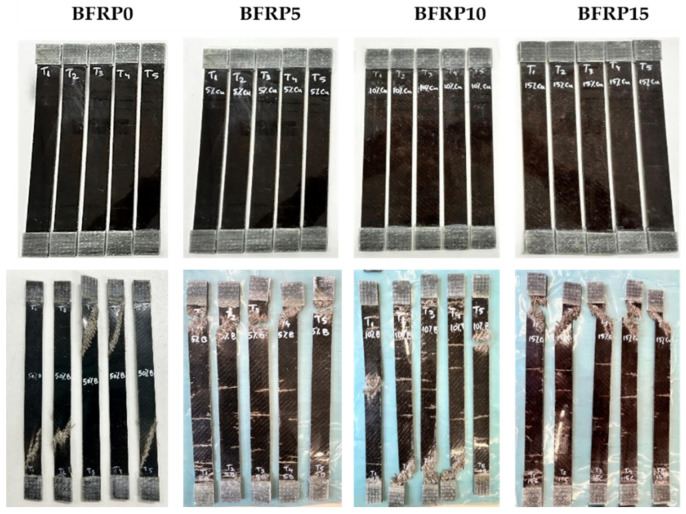
BFRP specimens before (row 1) and after tensile stress (row 2).

**Figure 8 polymers-17-00091-f008:**
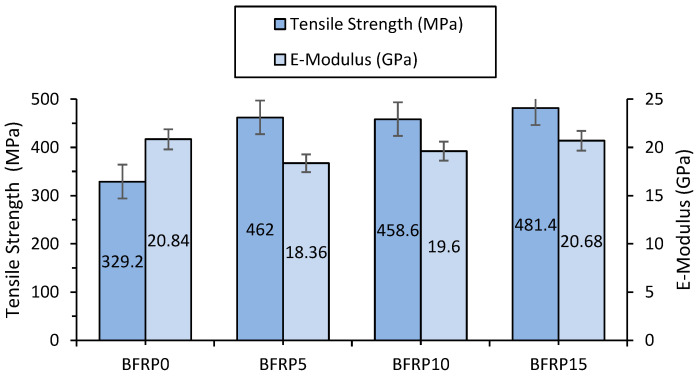
Tensile strength of BFRP laminates.

**Figure 9 polymers-17-00091-f009:**
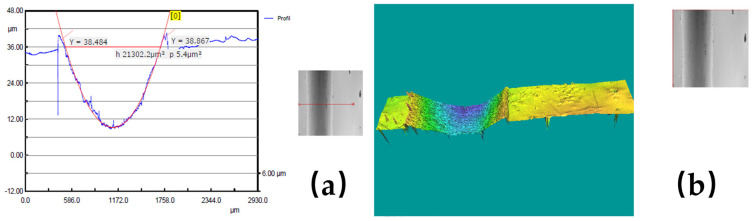
(**a**) Wear pattern for the BFRP5 disk, under test conditions: F_n_ = 20 N și v_1_ = 0.1 ms^−1^; (**b**) 3D optical profilometry of wear pattern.

**Figure 10 polymers-17-00091-f010:**
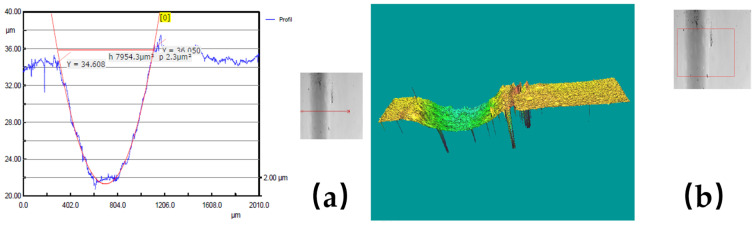
(**a**) Wear pattern for the BFRP10 disk under test conditions: F_n_ = 10 N și v_1_ = 0.1 ms^−1^; (**b**) 3D optical profilometry of wear pattern.

**Figure 11 polymers-17-00091-f011:**
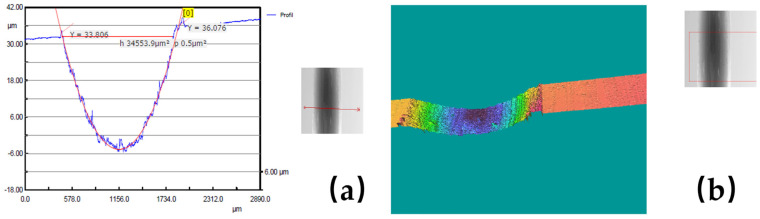
(**a**) Wear pattern for the BFRP15 disk under test conditions: F_n_ = 10 N și v_3_ = 0.36 ms^−1^; (**b**) 3D optical profilometry of wear pattern.

**Figure 12 polymers-17-00091-f012:**
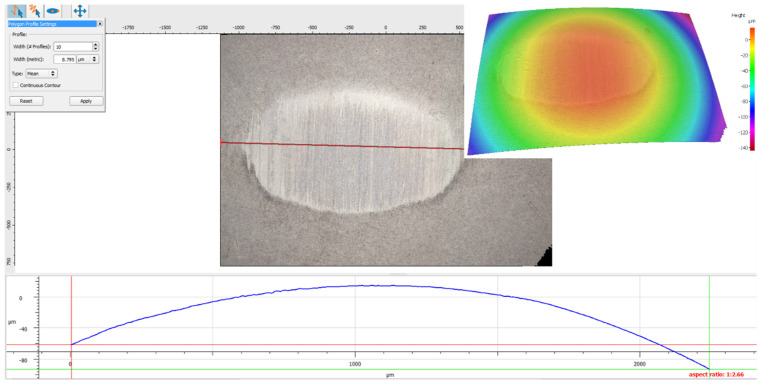
Wear marks of Chrome alloy steel ball under test conditions: F_n_ = 10 N și v_1_ = 0.1 ms^−1^.

**Figure 13 polymers-17-00091-f013:**
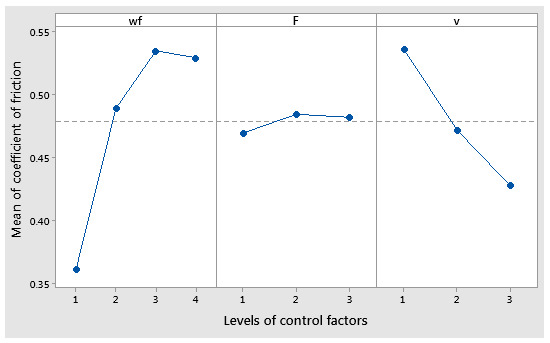
Main effects plot for coefficient of friction factor.

**Figure 14 polymers-17-00091-f014:**
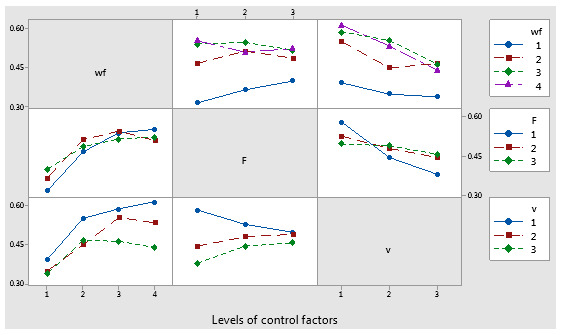
Interaction effects plot for coefficient of friction factor.

**Figure 15 polymers-17-00091-f015:**
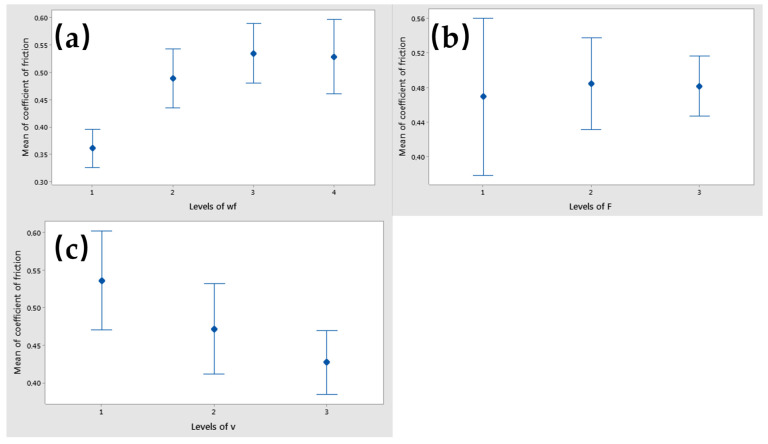
Interval plot of coefficient of friction factor with (**a**) wf_Cu; (**b**) F; and (**c**) v. Individual standard deviations were used to calculate the interval plot. Bars are standard errors of the mean.

**Figure 16 polymers-17-00091-f016:**
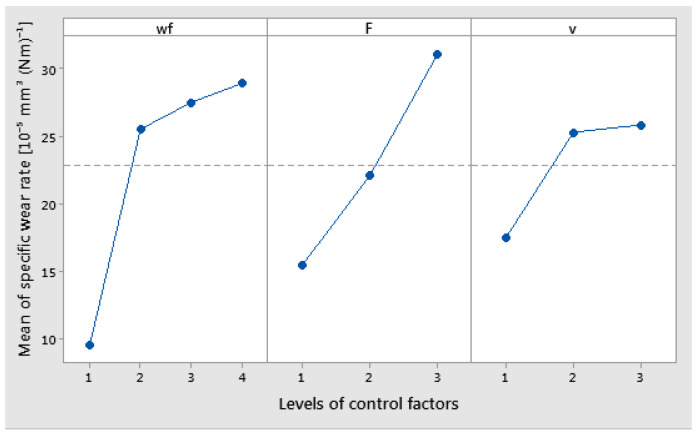
Main effects plot for coefficient of specific wear rate.

**Figure 17 polymers-17-00091-f017:**
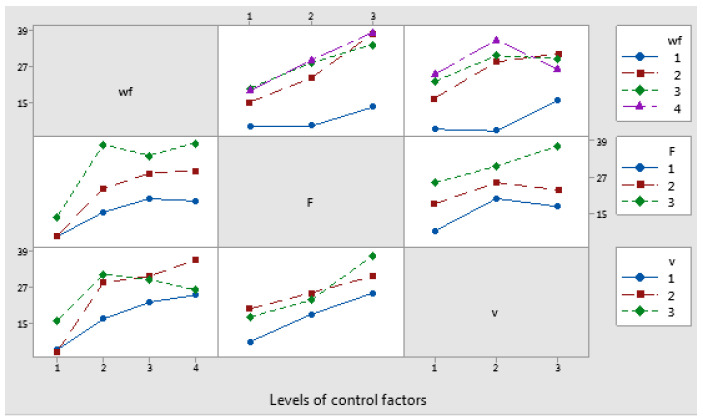
Interaction effects plot for specific wear rate factor.

**Figure 18 polymers-17-00091-f018:**
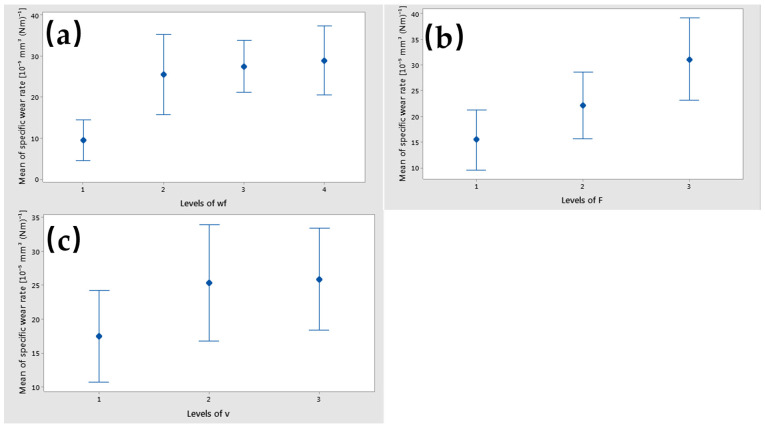
Interval plot of specific wear rate factor with (**a**) wf_Cu; (**b**) F; and (**c**) v. Individual standard deviations are used to calculate the interval plot. Bars are standard errors of the mean.

**Figure 19 polymers-17-00091-f019:**
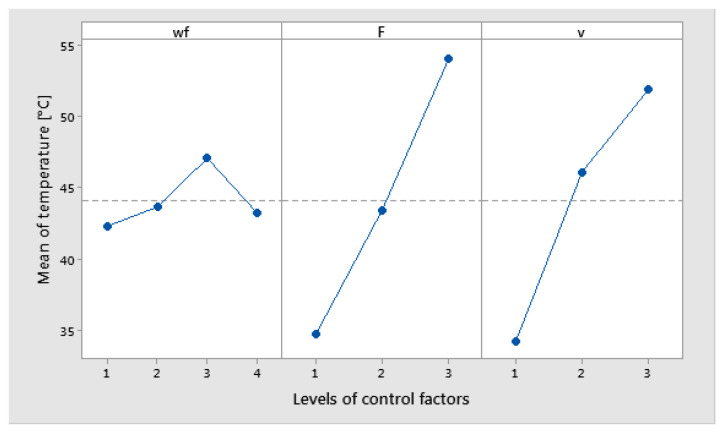
Main effects plot for temperature factor.

**Figure 20 polymers-17-00091-f020:**
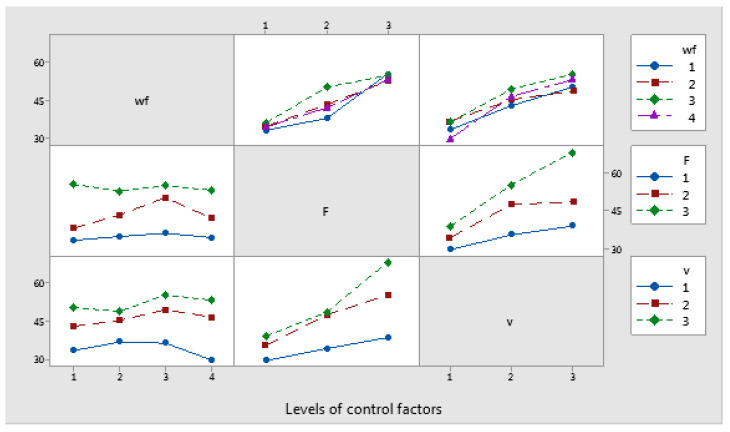
Interaction effects plot for temperature factor.

**Figure 21 polymers-17-00091-f021:**
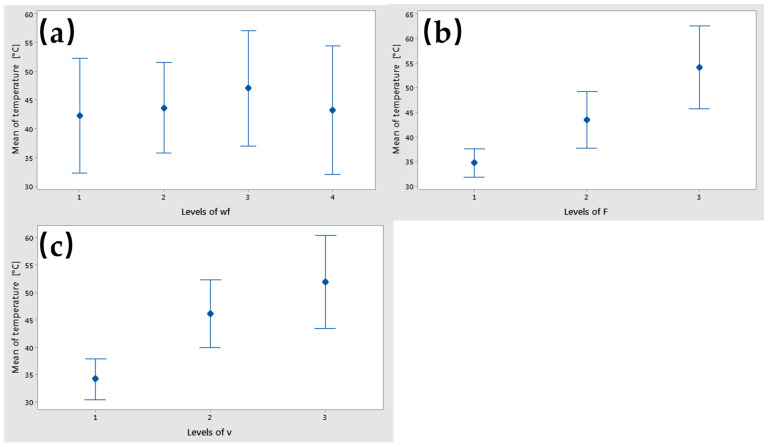
Interval plot of specific temperature factor with (**a**) wf_Cu; (**b**) F; and (**c**) v. Individual standard deviations are used to calculate the interval plot. Bars are standard errors of the mean.

**Figure 22 polymers-17-00091-f022:**
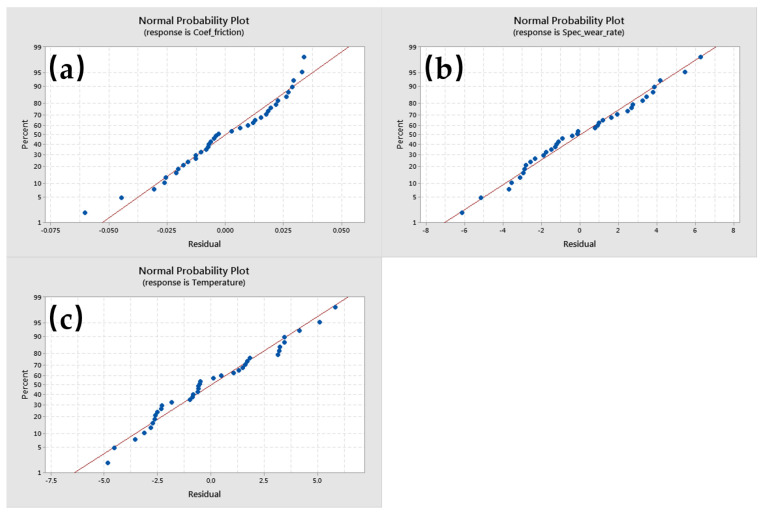
Normal probability plots of residuals for (**a**) coefficient of friction; (**b**) specific wear rate; and (**c**) temperature.

**Table 1 polymers-17-00091-t001:** Laminate design and composition.

Composite Laminates	Coding for Laminates	Basalt Fiber (wf.%)	Epoxy Resin (wf.%)	Copper Powder (wf.%)
Epoxy/basalt fiber reinforced polymer with 0 wf.% copper powder	BFRP0	50	50	0
Epoxy/basalt fiber reinforced polymer with 5 wf.% copper powder	BFRP5	50	45	5
Epoxy/basalt fiber reinforced polymer with 10 wf.% copper powder	BFRP10	50	40	10
Epoxy/basalt fiber reinforced polymer with 15 wf.% copper powder	BFRP15	50	35	15

**Table 2 polymers-17-00091-t002:** Chemical and mechanical properties of ball material.

Ball Type (12.7 mm)	Chemical Composition [%]	Mechanical Properties of Balls Used in the Test					
		Hardness (HRC) Scale	Compressive Strength (MPa)	Yield Strength (MPa)	Young’s Modulus (GPa)	Poisson’s Ratio	Roughness Ra (µm)
**52100 High quality carbon steel alloyed with chromium** **ρ = 7.81 g/cm^3^**	Fe: 96.5–97.3 C: 0.98–1.1 Si: 0.15–0.35 Cr: 1.4–1.6 Mn: 0.25–0.45 P and Si	54–58	2100–2200	2000	200	0.3	0.282–0.30

**Table 3 polymers-17-00091-t003:** Tribological testing parameters.

Parameters	Operating Conditions
Load (F)	10, 20, 30 N
Sliding speed (v)	0.1, 0.25, 0.36 m s^−1^
Rotational speed (rpm)	Max 215 (±3) rpm
Relative humidity RH% (RH)	45 (±5)%
Starting temperature (T_o_)	22 (±2) °C
Test duration (t)	120 min
Lubricating conditions	Dry friction
Disk/ball materials	Polymer composites reinforced with basalt fibers and copper powder/Chromium alloyed carbon steel balls 52100
Average roughness of the disk surface (R_a_ disk)	0.38 µm

**Table 4 polymers-17-00091-t004:** Flexural proprieties of 50% BFRP.

Specimen Type	Flexure Stress (MPa)	Flexure Strain (%)	E-Modulus (MPa)
BFRP0	133.23	1.29	10,313.04
BFRP5	157.34	1.57	10,011.46
BFRP10	140.28	1.49	9410.07
BFRP15	150.82	1.82	8287.20

**Table 5 polymers-17-00091-t005:** Tensile proprieties of 50% BFRP.

Specimen Type	Tensile Strength (MPa)	Tensile Strain at Tensile Strength (mm/mm)	E-Modulus (MPa)
BFRP0	329.2	0.0184	20,840
BFRP5	462	0.0274	18,360
BFRP10	458.6	0.0276	19,600
BFRP15	481.4	0.028	20,680

**Table 6 polymers-17-00091-t006:** Control factors and their levels for target analysis.

Targets	Cu Weight Fraction, wf%		Applied Load, F		Sliding Speed, v	
	Symbol	Value [%]	Symbol	Value [N]	Symbol	Value [m/s]
Coefficient of friction Specific wear rate Temperature	1	0	1	10	1	0.10
	2	5	2	20	2	0.25
	3	10	3	30	3	0.36
	4	15	-	-	-	-

**Table 7 polymers-17-00091-t007:** Experimental results for dry sliding wear.

Experimental Parameters			Optimizing Parameters			
Exp. No.	Applied Load F [N]	Sliding Speed v [m/s]	Copper Content Cu wf [%]	Specific Wear Rate K [10^−5^ mm^3^ (N m)^−^¹]	Coefficient of Friction (µ) Average of the Last 60 min	Temperature Average of the Last 60 min [°C]
**1**	10	0.1	0	3.247	0.36	30.87
**2**	10	0.1	5	32.133	0.61	29.96
**3**	10	0.1	10	11.0401	0.66	29.78
**4**	10	0.1	15	10.245	0.7	27.68
**5**	10	0.25	0	2.695	0.31	32.35
**6**	10	0.25	5	14.3962	0.35	33.06
**7**	10	0.25	10	55.707	0.56	38.45
**8**	10	0.25	15	34.4851	0.56	38.32
**9**	10	0.36	0	16.022	0.28	36.87
**10**	10	0.36	5	38.1892	0.44	41.54
**11**	10	0.36	10	19.68	0.4	40.25
**12**	10	0.36	15	13.3928	0.4	37.25
**13**	20	0.1	0	7.651	0.4	26.97
**14**	20	0.1	5	14.788	0.54	40.79
**15**	20	0.1	10	23.159	0.58	39.48
**16**	20	0.1	15	27.0731	0.59	30.02
**17**	20	0.25	0	5.8449	0.34	38.03
**18**	20	0.25	5	30.555	0.51	46.22
**19**	20	0.25	10	29.3781	0.56	58.04
**20**	20	0.25	15	26.1272	0.51	47.59
**21**	20	0.36	0	9.09	0.36	49.23
**22**	20	0.36	5	19.337	0.49	43.19
**23**	20	0.36	10	32.438	0.5	53.03
**24**	20	0.36	15	25.3018	0.43	48.74
**25**	30	0.1	0	8.693	0.42	42.87
**26**	30	0.1	5	23.333	0.5	39.8
**27**	30	0.1	10	32.3223	0.52	40.05
**28**	30	0.1	15	36.285	0.55	31.99
**29**	30	0.25	0	9.2219	0.4	58.63
**30**	30	0.25	5	40.812	0.49	56.77
**31**	30	0.25	10	33.765	0.54	51.95
**32**	30	0.25	15	39.0394	0.53	53.9
**33**	30	0.36	0	23.29	0.38	65
**34**	30	0.36	5	65.227	0.47	61.63
**35**	30	0.36	10	36.881	0.49	72.62
**36**	30	0.36	15	40.0741	0.49	73.60

**Table 8 polymers-17-00091-t008:** The percentage contribution ratio for coefficient of friction.

Source	DF	Adj SS	Adj MS	F-Value	*P*-Value	PC [%]
wf	3	0.176011	0.058670	38.80	<0.001	52.43
F	2	0.00155	0.000775	0.51	0.612	0.46
v	2	0.071217	0.035608	23.55	<0.001	21.21
wf * F	6	0.016739	0.002790	1.85	0.173	4.99
wf * v	6	0.020406	0.003401	2.25	0.109	6.08
F * v	4	0.031633	0.007908	5.23	0.011	9.42
Error	12	0.018144	0.001512			5.4
Total	35	0.3357				100

Note: “*” signifies the interaction between factors.

**Table 9 polymers-17-00091-t009:** The percentage contribution ratio for specific wear rate.

Source	DF	Adj SS	Adj MS	F-Value	*P*-Value	PC [%]
wf	3	2825.3	941.76	7.52	0.004	41.29
F	2	1057.3	528.67	4.22	0.041	27.94
v	2	573.1	286.54	2.29	0.144	9.94
wf * F	6	382	63.67	0.51	0.791	4.77
wf * v	6	695	115.84	0.93	0.511	7.45
F * v	4	343.5	85.89	0.69	0.615	2.55
Error	12	1502.3	125.19			6.06
Total	35	7378.7				100

Note: “*” signifies the interaction between factors.

**Table 10 polymers-17-00091-t010:** The percentage contribution ratio for temperature.

Source	DF	Adj SS	Adj MS	F-Value	*P*-Value	PC [%]
wf	3	116.7	38.90	1.75	0.210	2.19
F	2	2258	1129.02	50.76	<0.001	42.32
v	2	1959.7	979.86	44.06	<0.001	36.73
wf * F	6	139.2	23.20	1.04	0.445	2.61
wf * v	6	117.5	19.58	0.88	0.538	2.2
F * v	4	477.8	119.45	5.37	0.010	8.95
Error	12	266.9	22.24			5
Total	35	5335.8				100

Note: “*” signifies the interaction between factors.

## Data Availability

The original contributions presented in the study are included in the article; further inquiries can be directed to the corresponding author(s).

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
