# Peer review of "Enhanced Tribological and Mechanical Properties of Copper-Modified Basalt-Reinforced Epoxy Composites"

_polymers, 2025, doi:10.3390/polym17010091_

Round 1

Reviewer 1 Report

Comments and Suggestions for Authors

This paper gives a thorough study on the influence of copper powder on the mechanical and tribological properties of basalt fiber-reinforced polymer (BFRP) composites. This research is primarily important in the context of advances in the materials engineering industry, where demands are significantly increasing for lightweight and high-performance materials in various industrial applications. One major aspect of the research is the introduction of copper powder at different weight fractions (5, 10, and 15 percent) into the epoxy matrix. Results showed that because of the copper addition, composites possessed an improvement in tensile strength along with flexural modulus and ductility at higher copper concentrations. Tribological tests conducted through the pin-on-disc tribometer yielded a remarkable decrease in wear rates compared with the pure/untreated composites, thus indicating greater wear resistance of copper-modified BFRP composites compared to unmodified ones. There are many important areas for improvement and recommendation from the existing literature that can improve this article further:

* Exploring further the tribological mechanisms of basalt fiber-reinforced composites.

* A friction layer's formation and wear debris behavior is a major parameter in their performance in mechanical application areas. But currently, there are no good appreciations of these processes per Birleanu, quoting that "the mechanisms of friction layer formation have not-yet. experimental proof, would complement the article significantly on these mechanisms.

* Most importantly is the need for quantification of the mechanical properties of basalt fiber reinforced composites in the article. For example, it has been proven by nanoparticle study that even SiOâ‚‚ alone can increase tensile strength and ILSS significantly. In essence, the inclusion of such data will bring home to the articles the extent of improvements that modification techniques and other treatments could bring about in its claims.

* This article can develop the dimension regarding the bearing of different manufacturing methods on the mechanical properties of basalt fiber composite materials. Research shows that there is a ILSS value change depending on the technique of manufacturing. By discussing such findings, an insight may be gained in streamlining production processes.

* This article should show how surface treatments and fiber sizing can affect the mechanical properties of basalt fibers because fiber sizing optimization has been shown to bring a significant effect on mechanical properties, where sized fibers outperform unsized ones. This is relevant to practitioners aiming at specifics to enhance the real applications of basalt fiber composites.

* An extensive literature review on the tribological and mechanical characteristics of basalt fiber reinforced composites would certainly do the article well. Bringing in findings of various studies that study the wear behavior and mechanical enhancement of basalt fiber composites would, thus, give a wider perspective on the discussion and support the article's conclusions.

Author Response

Thank you for your detailed feedback and valuable suggestions, which we have carefully considered and incorporated into the revised manuscript. We have thoroughly reviewed the text to correct all typographical and grammatical errors, and the additions or replacements have been marked in red for clarity. To address the length concerns, we have condensed the introduction, abstract, and conclusions by eliminating repetitive or less relevant information while maintaining a clear focus on the study's key findings and their implications. The conclusions were rewritten to succinctly present the most significant outcomes and provide a future roadmap, avoiding repetition of details already discussed elsewhere in the manuscript. While the manuscript retains a substantial number of figures and tables, we believe these are essential for effectively conveying the data and supporting the conclusions, ensuring the study’s clarity and scientific contribution. We hope these revisions meet your expectations, and we remain open to any further suggestions to enhance the manuscript.

Find all our point-by-point responses to your comments and suggestions in the attached file.

Reviewer 2 Report

Comments and Suggestions for Authors

A manuscript entitled “Enhanced Tribological and Mechanical Properties of Copper-Modified Basalt-Reinforced Epoxy Composites” is very well written and structured by the authors. The manuscript may be accepted after incorporating minor modifications in the manuscript. The changes required in the manuscript is as follow:

·       Authors must incorporate the motivation/need in 2-3 sentences in abstract before explaining current work.

·       Introduction is too lengthy author try to concise the literature review section.

·       Application part is completely missing in introduction, author must add one paragraph related to applications of developed material.

·       Novelty must more highlighted in second last paragraph of introduction section.

·       Rectify this sentence there must be some error “The autoclave cicle were: 180 min at 1200 C, vaccum bag preassure -0.9 bars, internal preassure 4 bars in autoclave, cooling 60 min up to 300 C” in section 2.1

·       Author must add one table in which he compares their results with recently reported literature.

·       Authors must add relevant reasons for increase/decrease in the properties.

·       Authors needs to read manuscript twice before submitting revision. There are many typo errors and looks non-native English speaker write the manuscript.

·        Manuscript is very lengthy author needs to revisit the full manuscript thoroughly and remove the unnecessary and repetitive information.

·       Conclusion is too lengthy. Rewrite the conclusions precisely by removing repetition as discussed in abstract and discussion part. Only give the crux/ key outcomes having significance for scientific community along with future road map.

Author Response

(The authors gave the same response as above.)
